# DeTrack: In-model Latent Denoising Learning for Visual Object Tracking

**Xinyu Zhou**[1]    **Jinglun Li**[2]    **Lingyi Hong**[1]    **Kaixun Jiang**[2]    **Pinxue Guo**[2]

**Weifeng Ge**[1*]   **Wenqiang Zhang**[1,2*]

[1]Shanghai Key Lab of Intelligent Information Processing,
School of Computer Science, Fudan University, Shanghai, China
[2]Shanghai Engineering Research Center of AI & Robotics,
Academy for Engineering and Technology, Fudan University, Shanghai, China
zhouxinyu20@fudan.edu.cn, jingli960423@gmail.com, lyhong22@m.fudan.edu.cn,
kxjiang22@m.fudan.edu.cn, pxguo21@m.fudan.edu.cn,
weifeng.ge.ic@gmail.com,wqzhang@fudan.edu.cn

## Abstract

Previous visual object tracking methods employ image-feature regression models or coordinate autoregression models for bounding box prediction. Image-feature regression methods heavily depend on matching results and do not utilize positional prior, while the autoregressive approach can only be trained using bounding boxes available in the training set, potentially resulting in suboptimal performance during testing with unseen data. Inspired by the diffusion model, denoising learning enhances the model's robustness to unseen data. Therefore, We introduce noise to bounding boxes, generating noisy boxes for training, thus enhancing model robustness on testing data. We propose a new paradigm to formulate the visual object tracking problem as a denoising learning process. However, tracking algorithms are usually asked to run in real-time, directly applying the diffusion model to object tracking would severely impair tracking speed. Therefore, we decompose the denoising learning process into every denoising block within a model, not by running the model multiple times, and thus we summarize the proposed paradigm as an in-model latent denoising learning process. Specifically, we propose a denoising Vision Transformer (ViT), which is composed of multiple denoising blocks. In the denoising block, template and search embeddings are projected into every denoising block as conditions. A denoising block is responsible for removing the noise in a predicted bounding box, and multiple stacked denoising blocks cooperate to accomplish the whole denoising process. Subsequently, we utilize image features and trajectory information to refine the denoised bounding box. Besides, we also utilize trajectory memory and visual memory to improve tracking stability. Experimental results validate the effectiveness of our approach, achieving competitive performance on several challenging datasets. The proposed in-model latent denoising tracker achieve real-time speed, rendering denoising learning applicable in the visual object tracking community.

---

*corresponding author.

38th Conference on Neural Information Processing Systems (NeurIPS 2024).

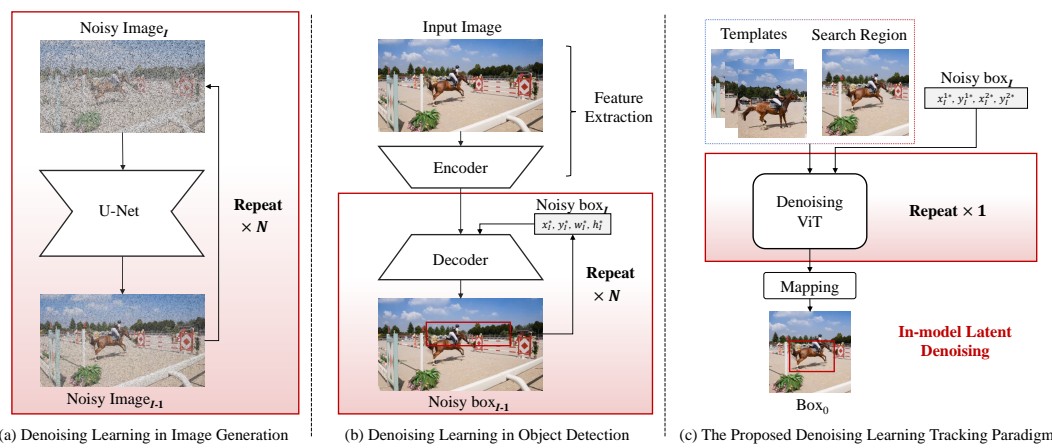

(a) Denoising Learning in Image Generation  (b) Denoising Learning in Object Detection  (c) The Proposed Denoising Learning Tracking Paradigm

Figure 1: **Difference of denoising learning paradigm. (a)** Diffusion model in image generation task. **(b)** Diffusion model in object detection task. **(c)** The proposed In-model latent denoising learning paradigm. The pink box indicates the denoising module. $\times N$ indicates denoising for $N$ times.

## 1   Introduction

Visual object tracking is a fundamental task in computer vision, which involves localizing and tracking a specific object in a video given its initial position. It finds broad applications in video understanding, surveillance, and robot navigation [45]. State-of-the-art approaches can be broadly categorized into two classes. The first class [17, 51, 41, 8, 19, 55, 29, 56] directly predicts the bounding box of the tracked target based on image features. The second class [6, 43], employs the coordinate autoregression framework.

While the mainstream methods have achieved prominent success, there are still certain issues to be addressed. Methods based on the image-feature regression framework rely heavily on the matching results between the template and the search region, which cannot utilize positional prior. Meanwhile, based on the autoregressive approach, it is necessary to utilize the bounding box from the previous frame to train the model, which can only utilize the existing bounding boxes in the training set. Therefore, during the testing phase, it may exhibit suboptimal performance for some unseen data.

The Diffusion model has achieved significant success in image generation task, allowing the generation of many images not seen in the training set[38]. Inspired by the Diffusion model[26], we add noise to bounding boxes during training stage. The noisy box can have arbitrary size and position, thereby enhancing the robustness of the model to unseen data during testing stage. As illustrated in Fig.1 (a) and (b), the diffusion model in image generation task requires multiple iterations of U-Net, while in object detection task, denoising is accomplished through multiple iterations of the decoder. However, tracking algorithms are usually asked to run in real-time, directly applying the diffusion model to object tracking would severely impair tracking speed.

Therefore, motivated by DAE[40], we propose a novel denoising learning paradigm (*DeTrack*) for visual object tracking that decomposes the denoising learning process in every denoising block with a tracking model. We use templates, search region, and noisy boxes as inputs. During the denoising process, we inject the template and search region as conditions to predict the noises in previous predictions. We repeatedly conduct the conditional denoising process and finally achieve accurate object location prediction. Specifically, as shown in Fig.1 (c), we propose a novel denoising ViT. We decompose a complete denoising process into several denoising blocks within ViT model and implement every denoising operation with a denoising block. Then the denosing learning process can be implemented in a single forward pass of the tracking model, which can reduce the computational cost drastically. To benefit from the in-context information[21, 16, 9, 28, 20, 22], we also put the previously predicted bounding boxes into a trajectory memory, and put the templates from previous frame into a visual memory. We use them as additional conditions to help locate objects more accurately.

Our contributions can be summarized as follows:

- We propose a novel in-model latent denoising learning paradigm for visual object tracking, which provides a new perspective for the research community. It decomposes the classical explicit denosing process into several denoising blocks and solves the problem with a tracking network in a single forward pass, which is valuable for real applications.

- We present a tracking model including a denoising ViT, comprised of multiple denoising blocks. The denoising process can be completed by progressively denoising through the denoising blocks within ViT. Furthermore, we construct a compound memory in the model that improve the tracking results using visual features and trajectory.

- Experimental results on several popular experiments, including AVisT, GOT-10k, LaSOT, and LaSOT$_{ext}$, demonstrate that the proposed method achieve competitive results.

.

## 2   Related Work

**Visual Object Tracking.** The existing visual object tracking methods can be broadly categorized into two main classes. The first class[1, 11, 48, 46, 8, 19, 41, 32, 31, 51, 10, 35, 44, 54] involves directly regressing the bounding box from image features, the second class [43, 6] treats the bounding box as four distinct tokens, employing an autoregressive model to sequentially predict these four tokens.

In the first class, deep neural networks are initially used to extract visual features, followed by the design of various prediction heads for regressing the bounding box. Since 2016, some prevalent methods have adopted a two-stream framework, employing siamese networks to separately extract visual features from the template and the search region. One type of prediction head [46, 32, 31, 53] uses a branch to predict the possible location of the target and other branches to predict the corresponding bounding box for that location. Another type of prediction head [10, 48, 17] consists of two branches that predict the coordinates of the top-left and bottom-right corners. Subsequently, OSTrack [51] introduces a one-stream tracking paradigm that combines feature extraction and feature fusion into a single step, achieving a new state-of-the-art performance. For the second class, SeqTrack [6] proposes transforming the bounding box into four tokens, predicting them sequentially in the order of $x, y, w$, and $h$. When predicting the bounding box, each box requires four passes through the decoder. Another autoregressive method, ARTrack [43], is similar to SeqTrack but differs in that it incorporates trajectory information in the input to enhance the model's awareness of trajectories.

**Denoising Learning.** DDPM [26] introduces denoising diffusion learning, which enhances the quality and diversity of generated images by adding noise to and denoising images. Subsequently, denoising learning has experienced explosive growth, being applied in various domains and achieving significant success. In the Super-Resolution field, SR3 [37] leverages DDPM for conditional image generation, employing a stochastic denoising process for super-resolution. Meanwhile, CDM [27] comprises a sequence of multiple diffusion models, each responsible for generating images with progressively higher resolutions. In video generation, the Flexible Diffusion Model (FDM) [23] utilizes a generative model designed for sampling arbitrary subsets of video frames, facilitated by a specialized architecture tailored for this purpose. The Residual Video Diffusion (RVD) model [50] employs an autoregressive, end-to-end optimized video diffusion model. In addition to generative tasks, denoising learning has also found extensive applications in discriminative task. DiffusionDet [4] applies the diffusion model to object detection, utilizing DDIM [38] for denoising. However, this approach still requires multiple passes through the decoder for denoising, impacting inference speed.

## 3   Method

In this section, we start by formulating the proposed tracking paradigm learned through denoising learning (Section 3.1). Next, we present our overall model architecture (Section 3.2), which includes a proposed denoising ViT, a box refining and mapping module, and a compound memory.

### 3.1   How to Formulate the Denoising Learning Tracking Paradigm?

**Image and Box Inputs.** We utilize both visual memory and the search region as conditional inputs $c$, while introducing noisy boxes $\mathbf{x}_I$ to predict the true position of the target, where $I$ represents the $I$-th state in the denoising process. The visual memory stores templates, which are cropped based on

previous frames. The search region is cropped based on the current frame and encompasses the area where the target may be present. In training stage, inspired by DDPM[26], we obtain noisy boxes $\mathbf{x}_I$ by adding Gaussian noise $\epsilon$ to the ground truth box $\mathbf{x}_0$:

$$\mathbf{x}_I = \sqrt{\bar{\alpha}}\mathbf{x}_0 + \epsilon\sqrt{1-\bar{\alpha}}, \epsilon \in \mathcal{N}(0, \mathbf{I}), \tag{1}$$

where $\bar{\alpha} = \prod_{j=0}^{T} \alpha_j$ and $\alpha_j = 1 - \beta_j$. $\beta_j \in (0, 1)$ is the variance schedule, $T$ is the time step.

**Optimization for Denoising Learning.** We take the visual memory and search region as conditional inputs $c$, and predict the true target position $\mathbf{x}_0$ from the noisy box $\mathbf{x}_I$, $p_\theta(\mathbf{x}_0|\mathbf{x}_I)$, where $\theta$ represents the neural network parameters. We aim to maximize the probability $p_\theta$ that the neural network predicts $\mathbf{x}_0$, enabling the model to predict the true target position:

$$\text{maximize}(p_\theta(\mathbf{x}_0|\mathbf{x}_I, c)). \tag{2}$$

To maximize $p_\theta$, we need to make the predicted $\mathbf{x}_0'$ by the network $f_\theta$ close to the ground truth $\mathbf{x}_0$:

$$\mathbf{x}_0' = f_\theta(c, \mathbf{x}_I),$$
$$\text{minimize}|\mathbf{x}_0' - \mathbf{x}_0|. \tag{3}$$

**Decomposes the Denoising Process into Multiple Denoising Block within a Model.** According to the principle of Markov, we can expand Equation 2 into a Markov chain:

$$p_\theta(\mathbf{x}_0|\mathbf{x}_I, c) = p(\mathbf{x}_I)\prod_{i=1}^{I} p_\theta(\mathbf{x}_{i-1}|\mathbf{x}_i, c) = p(\mathbf{x}_I)\prod_{i=\frac{I}{l}}^{I} p_\theta(\mathbf{x}_{i-\frac{I}{l}}|\mathbf{x}_i, c). \tag{4}$$

In the traditional Diffusion model[26], each step $p_\theta(\mathbf{x}_{i-1}|\mathbf{x}_i, c)$ is iteratively predicted using a neural network model $f_\theta$. However, our denoising paradigm decomposes the iterations of neural network into the iterations of denosing blocks within a neural network, $f_\theta = \{d_1, d_2, \cdots, d_l\}$, where each denoising block $d_l$ is responsible for predicting a state $p_\theta(\mathbf{x}_{i-\frac{I}{l}}|\mathbf{x}_i, c)$, where $l$ denotes the number of blocks. This allows our model to complete denoising with only a single forward pass of the tracking model.

**Discussion on the Differences from the Diffusion Model.** *The proposed denoising learning tracking paradigm is not a diffusion model.* (1) In the reverse denoising process of diffusion model, sampling a noise from a standard Gaussian distribution introduces randomness, making it more suitable for generating diverse images in image generation tasks. However, bounding boxes for visual object tracking are deterministic. Therefore, our proposed DeTrack does not involve a sampling process in reverse denoising process, making it more suitable for visual object tracking. (2) Each step of diffusion model is predicted recursively using a neural network. The proposed DeTrack predicts states using denosing blocks within a network (3) The diffusion model requires iterative prediction of neural network, whereas our method only requires a single forward pass through the network. Please refer to the Appendix A.1 for detailed analysis.

## 3.2 Model Architecture

**Inputs representation.** As show in Fig. 2, we use noisy bounding boxes as input and take visual memory and a search region as conditional inputs. Visual memory stores multiple templates. Specifically, gaussian noise is added to the ground truth bounding box to obtain a noisy bounding box $\{x_I^{1*}, y_I^{1*}, x_I^{2*}, y_I^{2*}\} \in \mathbb{R}^{4\times 1}$, where * denotes noise addition, while 1 and 2 respectively denote the upper left corner and lower right corner. Subsequently, the noisy box is mapped to a high-dimensional space by word embedding, resulting in noisy box embedding $\mathbf{x}_I \in \mathbb{R}^{4\times C}$. Additionally, we map templates and the search region to templates embedding $z \in \mathbb{R}^{N_z \times C}$ and search embedding $s \in \mathbb{R}^{N_s \times C}$ by a image embedding, where $N_z = n \times \frac{H_z}{16} \times \frac{W_z}{16}$, $N_s = \frac{H_s}{16} \times \frac{W_s}{16}$, $n$ denotes the number of templates, H and W represent the height and width of the image, respectively. For details, please refer to Model implement details in Section 4.1.

**Denoising ViT (In-model Latent Denoising).**

*ViT Transformer Block.* The specific transformer block structure is the same as the ViT transformer block[13]. Therefore, we only introduce integrating the features of templates and search region within the ViT block. Specifically, we perform attention on image embedding. We first obtain $q_s$ (search

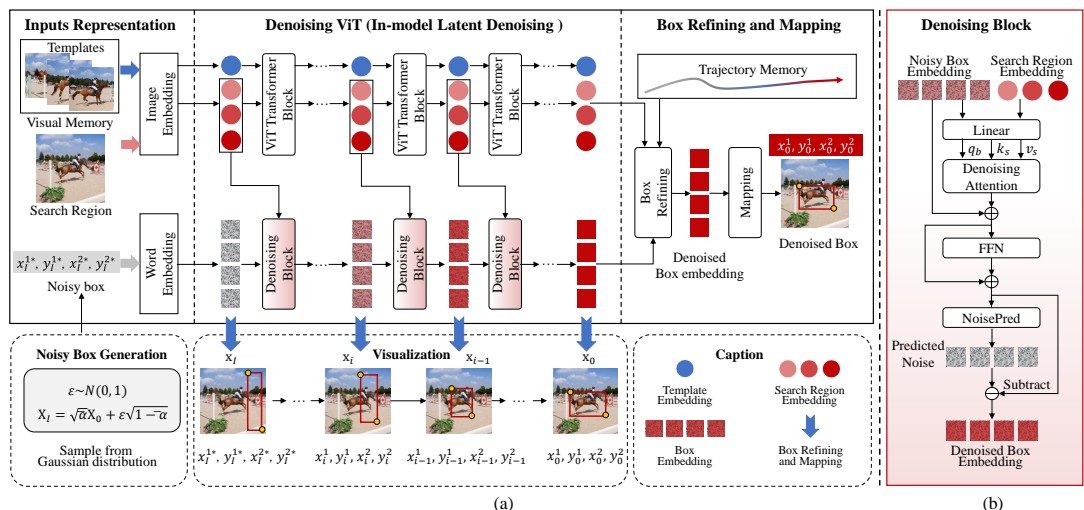

Figure 2: **The overview of model architecture. (a)** The model architecture comprises the input representation, the proposed Denoising ViT, and Box Refining and Mapping. It also includes Visual Memory and Trajectory Memory. **(b)** The proposed Denoising Block within Denoising ViT.

query), $q_z$(templates query), $k_s$(search key), $k_z$(templates key), $v_s$(search value) and $v_z$(templates value) through linear layer. The image attention is employed to interact and fuse image embedding:

$$\text{Attention}_{Image}(z, s) = \text{Softmax}(\frac{[q_s, q_z][k_s, k_z]}{\sqrt{d}}[v_s, v_z]), \tag{5}$$

where $[\cdot]$ denotes concatenation. $d$ is the dimensionality of the key.

***Denoising Block.*** As shown in Fig. 2, the input to the denoising block comprises the noisy box embedding and the search region embedding. These are passed through linear layers to obtain the $q_{\mathbf{x}_i}$(box query), $k_s$(search key), and $v_s$(search value). Subsequently, a denoising attention mechanism is employed for the **first time** of denoising:

$$\text{Attention}_{Denoising}(s, \mathbf{x}_i) = \text{Softmax}(\frac{q_{\mathbf{x}_i}k_s}{\sqrt{d}}v_s). \tag{6}$$

Then, we incorporate a Feedforward Neural Network (FFN) layer to enhance $\mathbf{x}_i^{'}$:

$$\mathbf{x}_i^{'} = \text{Attention}_{Denoising}(s, \mathbf{x}_i) + \mathbf{x}_i. \tag{7}$$

$$\mathbf{x}_i^{''} = \mathbf{x}_i^{'} + \text{FFN}(\mathbf{x}_i^{'}), \tag{8}$$

Finally, we use two linear layers to predict noise for the **second time** of denoising. Subtracting the noise from the box embedding yields the result after denoising through a NoisePred module:

$$\epsilon = \text{NoisePred}(\mathbf{x}_i^{''}) = \text{Linear}(\text{ReLu}(\text{Linear}(\mathbf{x}_i^{''}))),$$
$$\mathbf{x}_{i-\frac{I}{l}} = \mathbf{x}_i^{''} - \epsilon. \tag{9}$$

Denoising is performed through $l$ Denoising blocks. Ultimately, denoising is accomplished with a single forward pass of the denosing ViT, resulting in denoised box embedding $\mathbf{x}_0$:

$$\mathbf{x}_0 = \mathbf{x}_I - \sum_{j=1}^{l} \epsilon_j. \tag{10}$$

**Box Refining and Mapping.** As shown in Fig.3(a), we start by applying self-attention to the trajectory and denoised box embedding. We maintain that the current box embedding can only attend to its preceding box embedding by an attention mask in the self-attention, introducing temporal information. Subsequently, the output of self-attention is used as a query for cross-attention with the

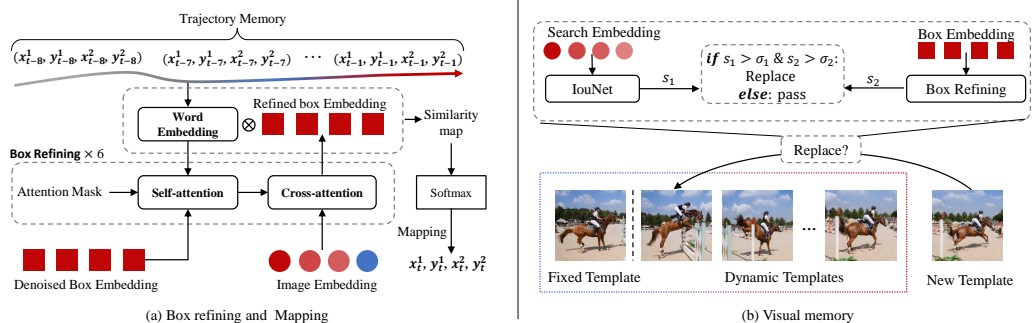

Figure 3: **Box refining and mapping and the updating of visual memory. (a)** Box refining and mapping introduces the trajectory memory to improve tracking performance. **(b)** Visual memory updating based on collaboratively decision including $s_1$ (IoU score) and $s_2$ (Softmax score).

image features. After undergoing six times of box refining, we compute the similarity between the refined box and word embedding, apply Softmax to obtain probabilities for different positions in the word embedding, and use the position with the highest probability as the bounding box, which is similar to ARTrack[43].

**Compound Memory.** We design a compound memory that includes both a visual memory and a trajectory memory. The visual memory enhances the model's ability to adapt to changes in the appearance of the target and the environment in the video. Besides, the trajectory memory enables the model to continue tracking the target even in the presence of occlusions or disappearances.

*Visual Memory.* As shown in Fig.3(b), our visual memory consists of dynamic templates and a fixed template. The first template of dynamic templates is discarded, and a new template is added. Directly updating the template can lead to cumulative errors. Therefore, we propose a collaborative updating mechanism. This involves inputting the search embedding extracted after Denoising ViT into IoUNet to obtain the corresponding IoU score $s_1$. Additionally, the Softmax score from Box Refining serves as a confidence value $s_2$. A collaborative decision on the quality of the new template frame is made based on two threshold values $\sigma_1$ and $\sigma_2$, determining whether updating.

*Trajectory Memory.* The proposed trajectory memory stores the boxes of the previous 7 frames, using a first-in-first-out (FIFO) approach when a new box needs to be stored. This results in a continuously updated trajectory box used for refining the denoised box. The trajectory memory can provide the model with prior positional information and target size, allowing accurate prediction of the bounding box even in cases of visual occlusion.

## 4 Experiments

### 4.1 Implementation Details

**Model implement details.** We design two variants of DeTrack with different resolutions as shown in Tab.1.

Table 1: The Floating-Point Operations per Second(FLOPs), and speed of the model variants.

| Model | Template Size | Search Region Size | Flops | Speed | Device |
|---|---|---|---|---|---|
| DeTrack$_{256}$ | $128 \times 128$ | $256 \times 256$ | 53.0G | 42FPS | RTX3090 |
| DeTrack$_{384}$ | $192 \times 192$ | $384 \times 384$ | 117.1G | 30FPS | RTX3090 |

Our denoising ViT adopts ViT-B [13] and utilizes MAE[25] for weight initialization, with a total of $l = 12$ denoising blocks. The box refining includes 6 transformer layers for self-attention and cross-attention. Additionally, we trained two models, namely DeTrack$_{256}$ and DeTrack$_{384}$. The template is cropped based on twice the size of the bounding box, while the search region is cropped based on four times (DeTrack$_{256}$) and five times (DeTrack$_{384}$) the size of the bounding box. To map

Table 2: State-of-the-art comparison on AVisT [36], GOT-10k [30], LaSOT [14] and LaSOT$_{ext}$ [15]. Where * denotes our model only trained on GOT-10k. The best results are highlighted in bold.

| Method | AVisT | | | GOT-10k* | | | LaSOT | | | LaSOT$_{ext}$ | | |
|---|---|---|---|---|---|---|---|---|---|---|---|---|
| | AUC | OP50 | OP75 | AO | SR$_{0.5}$ | SR$_{0.75}$ | AUC | P$_{Norm}$ | P | AUC | P$_{Norm}$ | P |
| SiamPRN++$_{255}$ [31] | 39.0 | 43.5 | 21.2 | 51.7 | 61.6 | 32.5 | 49.6 | 56.9 | 49.1 | 34.0 | 41.6 | 39.6 |
| DiMP$_{288}$ [2] | - | - | - | 61.1 | 71.7 | 49.2 | 56.9 | 65.0 | 56.7 | 39.2 | 47.6 | 45.1 |
| ATOM$_{288}$ [11] | 38.6 | 41.5 | 22.2 | - | - | - | 51.5 | 57.6 | 50.5 | 37.6 | 45.9 | 43.0 |
| PrDiMP$_{288}$ [12] | 43.3 | 48.0 | 28.7 | 63.4 | 73.8 | 54.3 | 59.8 | 68.8 | 60.8 | - | - | - |
| Ocean$_{255}$ [53] | 38.9 | 43.6 | 20.5 | 61.1 | 72.1 | 47.3 | 56.0 | 65.1 | 56.6 | - | - | - |
| Alpha-Refine$_{288}$ [48] | 49.6 | 55.7 | 38.2 | - | - | - | 65.3 | 73.2 | 68.0 | - | - | - |
| TransT$_{256}$ [7] | 49.0 | 56.4 | 37.2 | 67.1 | 76.8 | 60.9 | 64.9 | 73.8 | 69.0 | - | - | - |
| ToMP$_{288}$ [34] | 51.9 | 59.5 | 38.9 | - | - | - | 67.6 | 78.0 | 72.2 | 45.9 | - | - |
| DATT$_{256}$ [52] | - | - | - | 72.8 | 83.1 | 68.4 | 65.2 | 69.3 | 73.6 | - | - | - |
| TATrack$_{256}$ [24] | - | - | - | 73.0 | 83.3 | 68.5 | 68.1 | 77.2 | 72.2 | - | - | - |
| CTTrack$_{256}$ [39] | 56.3 | 66.1 | 44.8 | 71.3 | 80.7 | 70.3 | 67.8 | 77.8 | 74.0 | - | - | - |
| TMT$_{352}$ [42] | 48.1 | 55.3 | 33.8 | 67.1 | 77.7 | 58.3 | 63.9 | - | 61.4 | - | - | - |
| KeepTrack$_{352}$ [35] | 49.4 | 56.3 | 37.8 | - | - | - | 67.1 | 77.2 | 70.2 | 48.2 | - | - |
| STARK$_{320}$ [47] | 51.1 | 59.2 | 39.1 | 68.8 | 78.1 | 64.1 | 67.1 | 77.0 | - | - | - | - |
| AiATrack$_{320}$ [17] | - | - | - | 67.9 | 79.0 | | 69.6 | 80.0 | 63.2 | 47.7 | 55.6 | 55.4 |
| Mixformer$_{320}$ [10] | 56.5 | 66.3 | 45.1 | 70.7 | 80.0 | 67.8 | 69.2 | 78.7 | 74.7 | - | - | - |
| OSTrack$_{256}$ [51] | 54.2 | 63.2 | 42.2 | 71.0 | 80.4 | 68.2 | 69.1 | 78.7 | 75.2 | 47.4 | 57.3 | 53.3 |
| OSTrack$_{384}$ [51] | 57.7 | 67.3 | 48.3 | 73.7 | 83.2 | 70.8 | 71.1 | 81.1 | 77.6 | 50.5 | 61.3 | 57.6 |
| SwinTrack$_{224}$ [33] | - | - | - | 71.3 | 81.9 | 64.5 | 67.2 | 70.8 | - | 47.6 | 53.9 | - |
| SwinTrack$_{384}$ [33] | - | - | - | 72.4 | 80.5 | 67.8 | 71.3 | 76.5 | - | 49.1 | 55.6 | - |
| ROMTrack$_{256}$ [3] | 57.8 | 67.6 | 48.6 | 72.9 | 82.9 | 70.2 | 69.3 | 78.8 | 75.6 | - | - | - |
| ROMTrack$_{384}$ [3] | 59.1 | 68.7 | 50.5 | 74.2 | 84.3 | 72.4 | 71.4 | 81.4 | 78.2 | - | - | - |
| F-BDMTrack$_{256}$ [49] | - | - | - | 72.7 | 82.0 | 69.9 | 69.9 | 79.4 | 75.8 | 47.9 | 57.9 | 54.0 |
| F-BDMTrack$_{384}$ [49] | - | - | - | 75.4 | 84.3 | 72.9 | 72.0 | 81.5 | 77.7 | 50.8 | 61.3 | 57.8 |
| GRM$_{256}$ [18] | 54.5 | 63.1 | 45.2 | 73.4 | 82.9 | 70.4 | 69.9 | 79.3 | 75.8 | - | - | - |
| GRM$_{320}$ [18] | 55.2 | 64.2 | 46.8 | 73.4 | 82.9 | 70.5 | 69.9 | 79.3 | 75.8 | - | - | - |
| SeqTrack$_{256}$ [6] | 56.8 | 66.8 | 45.6 | 74.7 | 84.7 | 71.8 | 69.9 | 79.7 | 76.3 | 49.5 | 60.8 | 56.3 |
| SeqTrack$_{384}$ [6] | 57.8 | 67.4 | 48.0 | 74.8 | 81.9 | 72.2 | 71.5 | 81.8 | 77.8 | 50.5 | 61.6 | 57.5 |
| ARTrack$_{256}$ [43] | - | - | - | 73.5 | 82.2 | 70.9 | 70.4 | 79.5 | 76.6 | 46.4 | 56.5 | 52.3 |
| ARTrack$_{384}$ [43] | - | - | - | 75.5 | 84.3 | 74.3 | 72.6 | 81.7 | **79.1** | 51.9 | 62.0 | 58.5 |
| **DeTrack$_{256}$ (ours)** | 60.1 | **69.7** | **50.6** | 77.1 | 86.1 | 73.5 | 71.3 | 80.1 | 76.8 | 47.9 | 56.6 | 52.1 |
| **DeTrack$_{384}$ (ours)** | **60.2** | 69.1 | 50.2 | **77.9** | **86.5** | **74.9** | **72.9** | **81.7** | **79.1** | **53.6** | **64.4** | **60.4** |

boxes into a high-dimensional space, we utilize word embedding, similar to Pix2Seq [5], with the number of bins being 800 and 1200 for DeTrack$_{256}$ and DeTrack$_{384}$ respectively.

**Training.** Our experiments are conducted on Intel(R) Xeon(R) Gold 6326 CPU @ 2.90GHz with 252GB RAM and 8 NVIDIA GeForce RTX 3090 GPUs with 24GB memory. In the first stage, there is only visual memory, which randomly samples two frames from the video. The model is trained on full dataests (COCO, GOT-10k, TrackingNet, and LaSOT). A total of 240 epochs are trained, with the learning rate set to 8e-5 for the denoising ViT and 8e-6 for the box refining. The learning rate decreases by a factor of 10 at the 192-th epoch. In the second stage, trajectory memory is introduced to refine the box, and sequential training is adopted. Consecutive frames are sampled from the video, with each frame's prediction result stored in the trajectory memory and updated in a first-in-first-out manner. The training is conducted on three datasets excluding COCO. A total of 60 epochs are trained, with the learning rates decreasing to 4e-6 and 4e-7 for the denoising ViT and box refining, respectively. In the third stage, only IoUNet is trained while other parts are frozen. The learning rate is set to 1e-4, and a total of 40 epochs are trained, with a 10× learning rate decay at the 30-th epoch. For GOT-10k, the learning rate remains consistent with training on the full dataests. In the first stage, we train for 120 epochs, with a 10× decrease in learning rate at the 96-th epoch, followed by training for 25 epochs in the second stage. During the training on GOT-10k, IoUNet is not used. The loss functions is cross-entropy and SIoU, which is the same as ARTrack[43].

**Inference.** During the testing phase, we use the search region and template as image inputs and initialize the box with the previous box (predicted bounding box of t-1 frame). Additionally, the update interval of the visual memory is set to 5 for t <= 100, doubled every 100 frames until t = 500, and then remains 160. While testing on the GOT-10k dataset, the visual memory is updated directly. For other datasets, the IoU score and confidence score is applied to filter templates. The trajectory memory stores seven bounding boxes, updating with a frequency of every frame. Inference is conducted on an NVIDIA GeForce RTX 3090.

## 4.2 State-of-the-Art Comparisons

**AVisT**. The AVisT dataset, as described in [36], covers a broad spectrum of diverse and demanding situations, encompassing harsh weather conditions like thick fog, intense rainfall, and sandstorms. Our tracker demonstrates outstanding performance on AVisT [36], a dataset with extreme weather conditions and harsh environments. It outperforms SeqTrack$_{384}$ by 2.4% in AUC, substantiating our tracker's excellence in extreme environmental conditions.

**GOT-10k**. GOT-10k comprises a training dataset consisting of 10,000 videos and a testing dataset with 180 videos. There is no overlap between the training and test sets, necessitating trackers to demonstrate robust generalization capabilities towards unseen data. As shown in Tab. 2, our method demonstrates superior performance on the GOT-10k [30]. Our DeTrack$_{256}$ achieves a significant improvement in AUC compared to SeqTrack$_{256}$ [6], with increases of 3.0% and 2.4%, respectively. Our DeTrack$_{384}$ outperforms the state-of-the-art method ARTrack$_{384}$ by 2.4%. This is attributed to the non-overlapping nature of the training and testing sets in the GOT-10k dataset, indicating our method's strong performance on unseen data. The denoising learning paradigm has learned powerful denoising capabilities while facing with arbitrary positions and sizes of boxes.

**LaSOT**. LaSOT is benchmark designed for long-term tracking, featuring a test collection consisting of 280 videos. Our DeTrack256 achieves an AUC of 71.3%, exhibiting performance improvement compared to other methods based on 256 resolution. Additionally, our DeTrack384 also demonstrates state-of-the-art performance, validating the strong competitiveness of our approach in long-term dataset. This is attributed to our compound memory design, which leverages historical trajectory and appearance information to enhance the model's generalization ability on long-term dataset.

**LaSOT$_{ext}$**. LaSOT$_{ext}$ [15]is an extension of the LaSOT dataset, also categorized as a long-term tracking dataset. It comprises 150 video sequences and encompasses 15 object classes. Our De-Track384 shows significant improvements compared to other methods, with a 1.7% increase in AUC over SeqTrack384 and a 2.4% improvement in P$_{norm}$. This demonstrates the strong generalization capability of our approach even with extended data, particularly manifesting notable advantages in the accuracy of bounding box center point.

## 4.3 Ablation study on Denoising Learning

Table 3: Ablation study of denoising steps on GOT-10k. The best results are highlighted in bold.

| | step1 | step2 | step3 | step4 | step5 | step6 | step7 | step8 | step9 | step10 | step11 | step12 |
|---|---|---|---|---|---|---|---|---|---|---|---|---|
| AO | 1.1 | 1.6 | 4.8 | 7.5 | 12.5 | 21.4 | 33.1 | 52.3 | 65.7 | 70.2 | 74.8 | **77.1** |
| SR$_{0.5}$ | 0.1 | 0.2 | 1.2 | 2.8 | 8.0 | 17.9 | 34.1 | 57.6 | 74.7 | 78.7 | 83.7 | **86.1** |
| SR$_{0.75}$ | 0.0 | 0.0 | 0.2 | 0.8 | 2.9 | 8.0 | 17.8 | 39.1 | 56.9 | 64.6 | 70.5 | **73.5** |

**Influence of denoising steps.** We investigate the impact of the number of denoising iterations on the performance of the tracker. Our proposed In-model latent denoising consists of a total of 12 steps based on denosing blocks, requiring only a forward pass to complete denosing. As shown in Tab.3, the model's performance is nearly zero at the first and second denoising steps because the bounding boxes are still filled with noise. However, there is a significant qualitative improvement in model performance at the eighth denoising step, reaching its peak at the twelfth step. As shown in Fig.4, the results improve progressively step by step, consistent with Tab. 3.

**Analysis of denoising paradigm.** Although our method completes denoising with only a single forward pass through the tracking model, it can also be adapted to perform multiple forward passes, similar to traditional Diffusion model[26]. Therefore, we further analyze and compare the multiple

Table 4: Ablation study of denoising paradigm on GOT-10k. The best results are highlighted in bold.

| Denoising paradigm | Steps | AO | SR$_{0.5}$ | SR$_{0.75}$ | FLOPS | Speed |
|---|---|---|---|---|---|---|
| Multiple forward passes | 96 | 75.7 | 84.8 | 72.9 | 424.0G | 8FPS |
| Multiple forward passes | 48 | 75.7 | 84.6 | 72.5 | 212.0G | 12FPS |
| Multiple forward passes | 24 | 75.9 | 84.9 | 72.4 | 106.0G | 29FPS |
| Single forward pass | 12 | **77.1** | **86.1** | **73.5** | **53.0G** | **42FPS** |

Table 5: Ablation study of denoising block on GOT-10k. The best results are highlighted in bold.

| Denosing block | Denoising attention | NoisePred | AO | SR$_{0.5}$ | SR$_{0.75}$ | FLOPS |
|:---:|:---:|:---:|:---:|:---:|:---:|:---:|
| | ✓ | | 74.0 | 84.7 | 72.5 | 51.2G |
| ✓ | ✓ | | 75.1 | 84.1 | 72.0 | 52.7G |
| ✓ | ✓ | ✓ | **77.1** | **86.1** | **73.5** | **53.0G** |

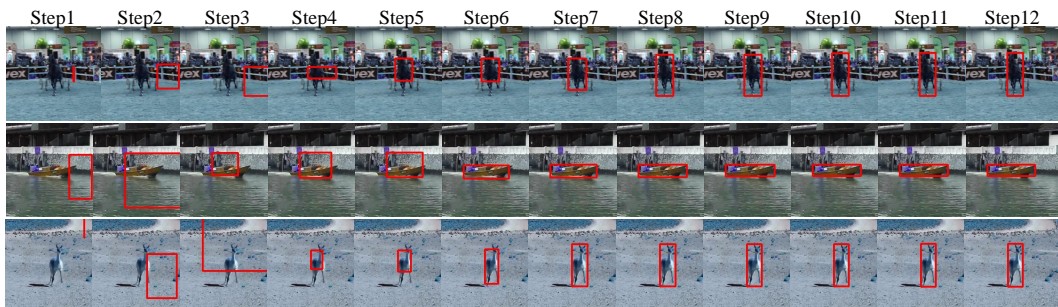

Figure 4: **Visualization of the denoising step GOT-10k.** The first row is the video GOT-10k-Test-000040, the second row is the video GOT-10k-Test-000003, and the third row is the video GOT-10k-Test-000051.

forward passes and single forward pass paradigms, as shown in Tab. 4. In DeTrack, the performance of multiple forward passes is not superior to that of single forward pass. Additionally, if denoising is performed similarly to traditional Diffusion models, the computational cost increases significantly. Single forward pass only requires 53.0G FLOPS and achieves a speed of 42 FPS, while multiple forward passes incurs exponentially higher computational costs with a linear decrease in speed.

**Analysis of the denoising block**. As shown in Tab.5, if there is no NoisePred module, AO will decrease by 2.0%, and SR$_{0.5}$ will decrease by 2.0%. This demonstrates that noise prediction and gradually subtracting noise are crucial for the model. Furthermore, removing denoising attention leads to further performance degradation, demonstrating that utilizing image features as conditional inputs can also assist in denoising. Moreover, the computational overhead of the denoising block increased by only 1.80G, owing to the fact that the box comprises merely 4 tokens. Thus, even with the addition of denoising attention and NoisePred, this remains a negligible computational burden.

### 4.4 Ablation study on Compound Memory

Because the memory mechanism is designed to address the challenge of dynamic changes in video, and considering the greater variety of environmental and appearance changes in long video datasets, we chose the LaSOT (long-term tracking dataset, averaging 2448 frames per video) to validate the effectiveness of our memory mechanism.

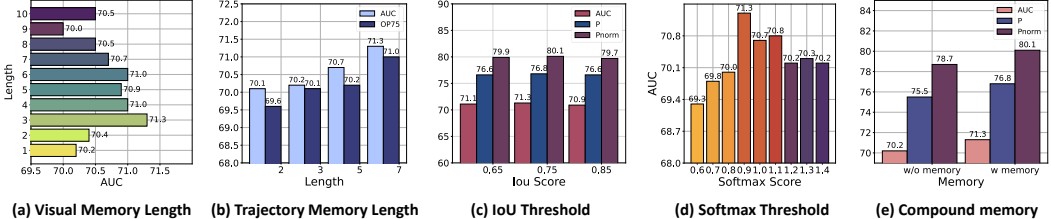

Figure 5: **Ablation study of memory on LaSOT.** (a) Different visual memory lengths; (b) Different trajectory memory lengths; (c) Different IoU thresholds are applied for template updates; (d) The influence of Softmax thresholds. (e) With or without compound memory.

**Exploration on the length of the visual memory and the trajectory memory**. We firstly explore the impact of different visual memory lengths. As shown in Fig.5 (a), when the length is only 1, the model's AUC is only 70.2. However, with an increase in memory length, performance gradually improves, reaching its peak at the 3-rd frame. Subsequently, performance declines. This is because when the memory is too short, the model cannot adapt to changes in the target and the environment.

Conversely, when the memory is too long, it stores incorrect information. Unlike visual memory, as shown in Fig. 5 (b), trajectory memory does not exhibit a trend of initially rising and then falling with an increase in stored boxes. The performance consistently improves as the number of boxes ranges from 1 to 7. As shown in Fig.5 (e), we also achieved a performance of 70.2 by removing all memories, which further confirms the effectiveness of our memory.

**Effects of IoU score and Softmax scorefor visual memory updating**. For the update of visual memory, we strive to avoid updating poor templates into our visual memory. This would lead to tracking drift. Therefore, as shown in Fig. 5(d) keeping IoU score fixed, we conduct an ablation study on different Softmax score values. The study found that an accuracy update can be achieved when the Softmax score is set to 0.9, obtaining 71.3% on AUC. As shown in Fig. 5(c) keeping Softmax score fixed, the best IoU score is 0.75. When the IoU score threshold is set to 0.85, it leads to a decrease in AUC. It is because the overly strict condition reduces the frequency of visual memory updates.

## 5  Limitation

Despite achieving real-time speed and competitive performance, our DeTrack still has certain limitations. Existing tracking methods struggle to recover the target when facing challenges such as object occlusion and out-of-view situations. Although our proposed trajectory memory can assist in target reacquisition after target loss in some cases, further improvements are needed to address challenges like object occlusion and out-of-view scenarios. We will investigate the challenges in these scenarios.

## 6  Conclusion

Traditional visual object tracking methods using image-feature regression or coordinate autoregression models faced limitations in handling positional priors and unseen data. Inspired by the diffusion model, we introduced denoising learning to enhance model robustness. Our approach, employing noisy bounding boxes for training, introduces a novel paradigm of denoising learning in object tracking. By decomposing the process into individual denoising blocks within our proposed denoising Vision Transformer (ViT), we achieved real-time performance while maintaining effectiveness. Experimental results demonstrate the efficacy of our method, showcasing competitive performance and rendering denoising learning applicable in the visual object tracking community.

**Acknowledgement** This work was supported by National Natural Science Foundation of China (No.62072112), National Natural Science Foundation of China under Grant Nos. 62106051 and the National Key R&D Program of China 2022YFC3601405.

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

# A  Appendix

Table 6: The differences between DDPM[26], DAE[40], and DeTrack[40].

| | DDPM | DAE | DeTrack |
|---|---|---|---|
| Noise Type | Gaussian noise | Gaussian noise | Gaussian noise |
| Input | Noisy image($x^*$) | Noisy image($x^*$) | Noisy box($b^*$) |
| Encoding | $z = f_\theta(x^*)$ | $z = f_\theta(x^*)$ | $z_{12}, z_{11} \cdots, z_1 = f_\theta(b^*)$ |
| Decoding | Noise $\epsilon_\theta = g_\theta(z)$ | image $x_\theta = g_\theta(z)$ | box $b_\theta = g_\theta(z_{12})$ |
| Optimization objective | $\epsilon - \epsilon_\theta$ | $x - x_\theta$ | $b - b_\theta$ |
| Inference | $x_{t-1} = \frac{1}{\sqrt{\alpha_t}}(x_t - \frac{\beta}{\sqrt{1-\bar{\alpha}_t}}\epsilon_\theta) + \sigma_t\epsilon$ | $x_\theta = g_\theta(z)$ | $b_\theta = g_\theta(z_{12})$ |

## A.1  The differences between DDPM, DAE, and DeTrack

. According to Tab.6, we compares and analyzes the differences between DDPM, DAE, and DeTrack in denoising learning, highlighting the advantages of the DeTrack model in visual object tracking. All three use Gaussian noise to simulate the noise characteristics of the input; however, they differ in input data, encoding methods, decoding methods, optimization objectives, and inference approaches. DDPM and DAE take noisy images as input ($x^*$), aiming to restore or generate high-quality images,

while DeTrack innovatively uses noisy bounding boxes ($b^*$) as input, making it more suitable for visual tracking tasks in complex backgrounds and scenarios with fast-moving objects.

In terms of encoding, DDPM and DAE use single-layer feature encoding to obtain $z = f_\theta(x)$; in contrast, DeTrack employs multi-layer feature encoding with layer-by-layer denoising within the model, resulting in multiple hidden states $z_i$ ($z_{12}, z_{11}, \ldots, z_1 = f_\theta(b)$). This layer-wise denoising approach retains and optimizes target feature information, enhancing robustness to bounding box noise. For decoding, DDPM's decoding target is to restore the noise $\epsilon_\theta = g_\theta(z)$, while DAE directly decodes the image $x_\theta = g_\theta(z)$. DeTrack, on the other hand, decodes to a denoised bounding box $b_\theta = g_\theta(z_{12})$, ensuring high-precision localization of the target bounding box.

Regarding the optimization objective, DDPM minimizes the error between generated noise and target noise ($\epsilon - \epsilon_\theta$), DAE minimizes the error between the denoised image and the original image ($x - x_\theta$), and DeTrack optimizes the error between the denoised bounding box and the original bounding box ($b - b_\theta$), making it more suitable for accurate visual target localization. For inference, DDPM uses a reverse diffusion process to progressively denoise and generate an image, while DAE and DeTrack directly generate denoised results in inference: DAE outputs the image $x_\theta = g_\theta(z)$, and DeTrack outputs the bounding box $b_\theta = g_\theta(z_{12})$.

Overall, DeTrack's multi-layer feature encoding with internal model denoising, specific decoding approach, and optimization objective enable it to exhibit higher robustness in noisy and complex backgrounds, making it well-suited for target tracking tasks in dynamic and complex scenarios.

Table 7: Comparison of noise prediction pattern on GOT-10k. The best results are highlighted in bold.

| | AO | $SR_{0.5}$ | $SR_{0.75}$ |
|---|---|---|---|
| Predicting the total noise | 75.2 | 84.1 | 71.4 |
| Predicting noise layer by layer | **77.1** | **86.1** | **73.5** |

## A.2 Comparison of noise prediction pattern

. According to 7, Predicting the total noise resulted in a decrease of 1.9 in AO, 2 in $SR_0.5$, and 2.1 in $SR_0.75$,compared to multiple noise predictions. We analyze that this is because predicting the total noise directly is more challenging than predicting it layer by layer. Layer-by-layer denoising allows the model to learn to filter out some noise at intermediate layers before arriving at the final result, rather than achieving it in one step.

## A.3 Analysis of Denoising Paradigm with ViT-Small

.Table 8 presents an ablation study on different denoising paradigms and step settings evaluated on the GOT-10k dataset with a Vit-Small backbone. We compare performance metrics such as Average Overlap (AO) and Success Rates at two different overlap thresholds ($SR_{0.5}$ and $SR_{0.75}$).

The results indicate that multiple forward passes generally yield better performance compared to a single forward pass. Specifically, a step count of 48 achieves the best AO, $SR_{0.5}$, and $SR_{0.75}$ values, with scores of 69.4, 78.5, and 63.4, respectively, highlighted in bold in Table 8. This suggests that while increasing the number of steps from 12 (single forward pass) to 48 improves performance, further increasing to 96 steps does not result in additional gains, possibly due to diminishing returns in iterative refinement or over-smoothing of features.

Table 8: Ablation study of denoising paradigm on GOT-10k (Vit-Small). The best results are highlighted in bold.

| Denoising paradigm | Steps | AO | $SR_{0.5}$ | $SR_{0.75}$ |
|---|---|---|---|---|
| Multiple forward passes | 96 | 68.9 | 78.2 | 63.2 |
| Multiple forward passes | 48 | **69.4** | **78.5** | **63.4** |
| Multiple forward passes | 24 | **69.4** | 78.0 | 63.0 |
| Single forward pass | 12 | 69.1 | 78.3 | 62.9 |

Notably, the AO metric remains at 69.4 for both 48 and 24 steps, although the success rates ($SR_{0.5}$ and $SR_{0.75}$) are slightly lower at 24 steps. This finding implies that 48 steps might strike a balance between computational efficiency and denoising effectiveness, providing optimal tracking performance without the need for excessive forward passes.

In summary, the experiments demonstrate that while iterative denoising is beneficial, there exists an optimal step count (48 in this case) that maximizes tracking accuracy. This demonstrates that our proposed DeTrack, when using ViT-Small as the backbone, can enhance tracking accuracy through a recursive denoising approach, similar to DDPM. However, this recursive denoising introduces a significant increase in computational complexity.

## A.4 Applying DiffusionDet to Tracking

Table 9: Comparison of Configurations between DiffusionTrack and DeTrack.

| Denoising paradigm | Encoder | Decoder |
| --- | --- | --- |
| DiffusionTrack | DeTrack Encoder | DiffusionDet Decoder |
| DeTrack | DeTrack Encoder | DeTrack Decoder |

DiffusionDet cannot be directly applied to object tracking, as it requires interaction between the template and search region in tracking. Therefore, as shown in Tab. 10, we use the Encoder from DeTrack, which enables this interaction, as the encoder for DiffusionDet. The decoder is taken from DiffusionDet. We call this model DiffusionTracking, and it uses a resolution of 384x384. For fairness, the learning rate, number of epochs, weight decay, and other training parameters are kept consistent.

Table 10: Performance Comparison on GOT-10k between DiffusionTrack and DeTrack. The best results are highlighted in bold.

| Denoising paradigm | Step | AO | $SR_{0.5}$ | $SR_{0.75}$ | FLOPS |
| --- | --- | --- | --- | --- | --- |
| DiffusionTrack | 1 | 71.8 | 81.0 | 69.6 | 120.0G |
| DiffusionTrack | 2 | 72.1 | 81.1 | 70.0 | 123.4G |
| DiffusionTrack | 4 | 71.9 | 81.1 | 69.3 | 133.5G |
| DiffusionTrack | 8 | 73.5 | 82.9 | 71.2 | 147.2G |
| DiffusionTrack | 12 | 72.5 | 81.9 | 70.2 | 162.7G |
| DeTrack | 12 | **77.9** | **86.5** | **74.9** | 119.0G |

## A.5 Comparison on GOT-10k between DiffusionTrack and DeTrack

The performance comparison on GOT-10k dataset between DiffusionTrack and DeTrack demonstrates notable differences in tracking accuracy and computational efficiency across various denoising steps. For DiffusionTrack, the tracking performance generally improves as the step count increases, reaching its peak with 8 steps, where the Average Overlap (AO) is 73.5%, $SR_{0.5}$ is 82.9%, and $SR_{0.75}$ is 71.2%. However, this improvement comes at the cost of increased computational requirements, with the FLOPS reaching 147.2G at 8 steps and 162.7G at 12 steps.

In contrast, DeTrack, tested with 12 steps, achieves the highest performance overall, with an AO of 77.9%, $SR_{0.5}$ of 86.5%, and $SR_{0.75}$ of 74.9%, surpassing all DiffusionTrack configurations. DeTrack also maintains lower computational complexity with 119.0G FLOPS, suggesting a more optimal balance of tracking accuracy and efficiency. This analysis indicates that while DiffusionTrack benefits from increased steps in tracking performance, DeTrack achieves superior results both in accuracy and computational efficiency.

