# OpenReview forum: "DeTrack: In-model Latent Denoising Learning for Visual Object Tracking"
_NeurIPS.cc/2024/Conference — NeurIPS 2024 poster_

### Official Review · Reviewer_b9Bg · 2024-07-11

**Soundness:** 3
**Presentation:** 3
**Contribution:** 3
**Rating:** 6
**Confidence:** 5

**Summary:**

The paper introduces bounding box denoising to visual object tracking. To make it more suitable for tracking, the paper devises an in-model denoising pipeline that reformulate the repetitive sampling process as multiple ViT blocks in one forward pass. It also proposes a compound memory strategy to assist long-term tracking with visual feature and trajectory cues. The paper provides extensive results to validate the efficacy of its designs. The proposed method, named DeTrack, achieves competitive performance across multiple datasets.

**Strengths:**

Q1) Different from previous image-feature regression and autoregressive methods, the idea of denoising learning for tracking is novel and well-motivated, and the proposed pipeline is streamlined and promising to lead a new tracking paradigm.

Q2) Convincing ablations and intuitive visualizations are provided to help understanding the mechanism of the proposed method and demonstrate its effectiveness.

Q3) The proposed tracker achieves a satisfactory performance gain on several benchmarks in a real-time running speed.

Q4) Implementation details are specified to ensure the reproducibility of the proposed method.

**Weaknesses:**

W1) It is unfair to claim that the pipeline of DiffusionDet (Fig. 1(b)) is inefficient for tracking. The advantage compared to the former pipeline should be explained more meticulously due to the following reasons:

- Frist, for Fig. 1(b), the computation efficiency mainly depends on the capacity of each block instead of the pipeline. In fact, the decoder block in DiffusionDet could be more lightweight than the ViT block in this paper.

- Second, the recycling process of DiffusionDet equals to cascading several denosing blocks that share weights. Therefore, it has no difference in terms of architecture compared to the proposed pipeline when the denoising step is the same.

- Third, as indicated in Table 3, the proposed pipeline has the potential to further improve its performance by using more denoising steps. However, while DiffusionDet allows inferring with an arbitrary number of steps, the proposed pipeline requires retraining more blocks and fails to benefit from such flexibility.

W2) The state-of-the-art comparison on TrackingNet, which is a representative benchmark in tracking, is missing. It should be easy to evaluate the proposed tracker on TrackingNet since the training configuration has been already aligned with other baselines. It would be great if the performance can be reported.

W3) Some minor issues in writing:

- The records in Table 2 needs to be double checked. For example, the proposed tracker does not achieve the best performance in $\text{P}_{\text{Norm}}$ on LaSOT, but it is marked in **bold**.

- When introducing the motivation, the authors claim that diffusion is more robust to unseen data. Though the claim is generally true, but it is expected to cite some related references accordingly.

- In Line 252, it is a little bit confusing to say "reaching its peak at the twelfth step" as there is no step13. Words like "perform best" could be more appropriate here.

**Questions:**

Q1) Have the authors ever applied the pipeline of DiffusionDet directly to tracking? If practical, how is its performance?

Q2) From Fig. 4, it seems that the bounding box is randomly initialized. Will the randomness affect the tracking performance much between different runs? While this is not a limitation, I am wondering if starting from a fixed anchor (whether calculated by statistics or learned as Sparse R-CNN) would help. Is it a choice that the authors have already considered?

**Limitations:**

The authors have discussed the limitations.

---

> ### Author Rebuttal · Authors · 2024-08-07
>
> We are grateful for the reviewer's acknowledgment of our work. We have taken the time to thoroughly address your questions and have included detailed responses to clarify any uncertainties.
> >***Q1**.Have the authors ever applied the pipeline of DiffusionDet directly to tracking? If practical, how is its performance?*
>
> - **DiffusionDet cannot be directly applied to object tracking, as it requires interaction between the template and search region in tracking**. Therefore, we use the Encoder from DeTrack, which enables this interaction, as the encoder for DiffusionDet. The decoder is taken from DiffusionDet. We call this model **DiffusionTracking**, and it uses a resolution of 384x384. For fairness, the learning rate, number of epochs, weight decay, and other training parameters are kept consistent.
>     The specific configuration is shown in the table below:
>
>   | **Denoising paradigm** | Encoder| Decoder|
>    |-|-|-|
>    |DiffusionTracking|DeTrack Encoder| DiffusionDet Decoder |
>    |DeTrack|DeTrack Encoder|DeTrack Decoder|
>  - DiffusionDet uses the DiffusionDet method to achieve tracking through iterative denoising. As shown in the table below, with an increase in the number of iterations, the computation significantly increases (120G -> 147G) and the speed significantly decreases (27fps -> 14fps). DeTrack holds a significant advantage over DiffusionTracking in terms of speed and performance, achieving 3-5 points higher on the GOT-10k dataset. It also scores 3 to 7 points higher on AVisT, Lasot, and LaSOT_ext. DeTrack achieves real-time performance (30fps), whereas DiffusionTracking only reaches 14fps. **The primary reason for this is that DeTrack is specifically designed for object tracking, while DiffusionTracking, derived from DiffusionDet, is more suited for object detection.**
> - |-|-|-|GOT-10k|-|-|-|
> |-|-|-|-|-|-|-|
> |**Denoising paradigm**|**Steps**|**AO**|**SR$_{0.5}$**|**SR$_{0.75}$**|**Flops**|**Speed**|
> |DiffusionTracking|1|71.8|81.0|69.6|120.0G|27fps|
> |DiffusionTracking|2|72.1|81.1|70.0|123.4G| 25fps |
> |DiffusionTracking|4|71.9|81.1|69.3|133.5|16fps|
> |DiffusionTracking|8|73.5|82.9|71.2|147.2G|14fps|
> |DiffusionTracking|12|72.5|81.9|70.2|**162.7G**|10fps|
> |**DeTrack**|12|**77.9**|**86.5**|**74.9**|119.0G|**30fps**|
>
> - |-|-|---|AVisT|---|---|TrackingNet|---|---|LaSOT|---|---|LaSOT$_{ext}$|---|Flops|Speed|
> |-|-|-|-|-|-|-|-|-|-|-|-|-|-|-|-|
> |**Denoising paradigm**|Steps|AUC|OP50|OP75|AUC|Pnorm|P|AUC|Pnorm|P|AUC|Pnorm|P|Flops|Speed|
> |DiffusionTracking|8|54.3|63.5|45.8|83.7|88.6|82.7|69.2|78.3|75.3|46.5|57.0|52.0|**147.1G**|14fps|
> |**DeTrack**|12|**60.2**|**69.1**|**50.2**|***84.5**|**89.1**|**83.6**|**72.9**|**81.7**|**79.1**|**53.6**|**64.4**|**50.4**|119.0G|**30fps**|
>
> >***Q2**.the computation efficiency mainly depends on the capacity of each block instead of the pipeline. In fact, the decoder block in DiffusionDet could be more lightweight than the ViT block in this paper.*
> - As shown in the table below, the DiffusionDet decoder has 22.6 times more computation than the DeTrack denoising block (3.4G vs 0.15G). This is because the DiffusionDet decoder contains 6 transformer layers and a series of fully connected layers.
> - |**Denoisingparadigm**|Step|Flops|
> |-|-|-|
> |DiffusionTracking Decoder|1|3.40G|
> |DiffusionTracking Decoder|8|27.20G|
> |DeTrack Denoising Block|1|0.15G|
> |DeTrack Denoising Block|12|1.80G|
>
> - If we make the DiffusionDet decoder more lightweight, it would lead to a performance drop. The decoder is crucial for both object detection and tracking as it helps us locate the target more accurately. To validate this, we reduced the computation of the DiffusionDet decoder, which resulted in a significant performance drop, as shown in the table below:
> - ||||-| GOT-10k|-||||
> |-|-|-|-|-|-|-|-|-|
> |**Denoisingparadigm**|**Steps**|**Decoder Flops**|**Number of Decoder TransformerLayer**|**AO**|**SR$_{0.5}$**|**SR$_{0.75}$**|**Flops**|**Speed**|
> |DiffusionTracking|8|0.56G|1|65.9|74.6|59.9|121.1G|26fps|
> |DiffusionTracking|8|3.4G|6|73.5|82.9|71.2|**147.2G**|14fps|
> |**DeTrack**|12|2.4G|6|**77.9**|**86.5**|**74.9**|119.0G|**30fps**|
>
> - When we reduced the DiffusionTracking decoder from 6 layers to 1 layer, the AO dropped by 7.6 points, and the SR0.75 dropped by 11.3 points.
>
> >***Q4**. It has no difference in terms of architecture compared to the proposed pipeline when the denoising step is the same.*
> - **Although DeTrack and DiffusionDet can be considered equivalent in terms of denoising steps in some form, this is not our primary contribution.** Our contribution lies in integrating the denoising process into the Vision Transformer (ViT), allowing for feature extraction, feature interaction, and denoising to be performed simultaneously. This approach enables DeTrack to complete denoising in a single forward pass, significantly reducing the computational load of the tracker.
>
> - In contrast, DiffusionTrack, derived from DiffusionDet, requires denoising after feature extraction and interaction, utilizing a Decoder. **While the reviewer suggests that the Decoder could be made more lightweight, the performance impact of the Decoder on object tracking is substantial.** A reduction in the Decoder's complexity results in a noticeable performance decline. As described in Table **Q2**, when the Decoder's computational cost is reduced from 3.4G to 0.56G, the AO drops by 7.6 points and SR$_{0.75}$ drops by 11.3 points. If the computational cost were to match the 0.15G of DeTrack's denoising block, performance might degrade further.

---

> > ### Comment · Reviewer_b9Bg · 2024-08-09
> >
> > Dear authors,
> >
> > Thanks for your detailed responses and adequate experiments, which have addressed most concerns of mine. After reading the rebuttal and comments from other reviewers, I have some follow-up questions:
> > - What is the DeTrack Encoder in the experiment for Q1? How many interaction blocks are used in it?
> > - In authors' response to Q6, the previous box is used as the input, typically centered in the search region with a similar bounding box size to the object in the current frame. That raises another question. From the visualization in Fig. 4, if the denoising starts from a reliable prior, why are the bounding boxes almost random after the first block? It appears that the first block serves as an adverse step in the denoising process that destroys the prior information.
> > - As other reviewers have suggested, there are some confusing expressions in the descriptions of denoising. While the authors have highlighted the differences from the Diffusion Model on Page 4, I recommend clarifying these differences earlier in the paper. In addition, I hope the discussions and experiments in the rebuttal can be incorporated as supplementaries of revision, which would be quite helpful for readers.

---

> > > ### Author Response · Authors · 2024-08-09
> > >
> > > Thank you for your valuable feedback. Please allow me to clarify further to alleviate your concerns.
> > > > ***Q1**. What is the DeTrack Encoder in the experiment for Q1? How many interaction blocks are used in it?*
> > > - DeTrack uses ViT-B (12 blocks) along with an additional 3 Transformer blocks as our encoder for feature interaction and feature extraction. This is a one-stream architecture, which allows for both feature extraction and feature interaction to be performed simultaneously.
> > >
> > > > ***Q2**. From the visualization in Fig. 4, if the denoising starts from a reliable prior, why are the bounding boxes almost random after the first block? It appears that the first block serves as an adverse step in the denoising process that destroys the prior information.*
> > > - We sincerely apologize for the confusion caused. To demonstrate the denoising effect, the bounding boxes in Figure 4 were visualized by sampling from Gaussian noise, which introduced an element of randomness. We will provide a detailed explanation of this in the revised version to enhance the quality of the manuscript.
> > >
> > > > ***Q3**.  I recommend clarifying these differences earlier in the paper. In addition, I hope the discussions and experiments in the rebuttal can be incorporated as supplementaries of revision, which would be quite helpful for readers.*
> > > - Thank you very much for your suggestions. Based on your feedback, we  will introduce the table ( Disccusion with reviewer wkvN) discussing the differences and similarities between DDPM, DeTrack, and DAE at an earlier section.
> > > - Additionally, as per your recommendation, we will include the disccusion and experimental analysis from the rebuttal in the supplementary materials to aid readers' understanding.
> > >
> > > Once again, thank you for your valuable suggestions.

---

> > > > ### Comment · Reviewer_b9Bg · 2024-08-09
> > > >
> > > > Dear authors,
> > > >
> > > > Thanks for your prompt reply. It is a good paper that should be recommended. I will maintain my initial rating as I think it is high enough.

---

> > > > > ### Author Response · Authors · 2024-08-09
> > > > >
> > > > > Thank you for your response, and we sincerely appreciate your recognition of our work.

---

> ### Author Response · Authors · 2024-08-07
> **Further response serve to adress reiviewer's concerns**
>
> >***Q5**.while DiffusionDet allows inferring with an arbitrary number of steps, the proposed pipeline requires retraining more blocks and fails to benefit from such flexibility.*
> - We acknowledge that increasing the denoising steps would require retraining the network. However, our design prioritizes efficiency and targeted performance. By carefully optimizing the existing 12 denoising modules, we ensure that the model is highly efficient with the predetermined number of denoising steps.
> - While DiffusionDet offers greater flexibility with its iterative approach, our method provides a computationally efficient alternative, particularly suitable for resource-constrained scenarios.
> - Overall, our approach strikes a balance between computational efficiency and flexibility, making it suitable for specific use cases where reducing the number of forward passes is crucial.
>
> >***Q6**. Will the randomness affect the tracking performance much between different runs? While this is not a limitation, I am wondering if starting from a fixed anchor would help. Is it a choice that the authors have already considered?*
>
> Due to the randomness introduced by Gaussian noise, different runs can indeed result in varying performance. Therefore, during the testing phase, we have used a fixed box (previous box). The previous box provides a prior on the target's position, leading to the highest performance.
> - |-|GOT-10k|-|-|
> |-|-|-|-|
> |**Boxtype**|**AO**|**SR$_{0.5}$**|**SR$_{0.75}$**|
> |Noisy box(First run)|74.8|83.3|71.0|
> |Noisy box(Second run)|75.6|84.5|73.4|
> |Previous box(Fixed anchor)|**77.9**|**86.5**|**74.9**|
>
> >***Q7**. It would be great if the performance can be reported.*
> - Based on your suggestion, we tested the TrackingNet dataset, and the results are shown in Table **Q1**. We will report these results in the revised version of the paper. Thank you very much for your input; it has been extremely helpful in improving our manuscript.
>
> >***Q8**. Some minor issues in writing*
> - Thank you for your valuable feedback. We appreciate your comments on the minor issues in writing. We have carefully reviewed the manuscript and addressed these issues to improve the clarity and readability of our paper.

---

### Official Review · Reviewer_wkvN · 2024-07-11

**Soundness:** 3
**Presentation:** 2
**Contribution:** 4
**Rating:** 7
**Confidence:** 5

**Summary:**

The paper addresses the task of visual object tracking. The main idea is to pose tracking as a latent denoising problem where the bounding box from the previous frame is denoised to obtain the current frame box. The authors use a ViT to obtain embeddings using the search regions as well as a set of template images as input. The embeddings obtained from each ViT block serves as the conditioning input to a denoiser block that iteratively denoises the noise bounding box estimate. The final box is further refined by a block which takes the trajectory (previous 7 estimated bounding boxes) as input. The authors maintain a dynamic memory of previous templates, based on the reliability of tracking. The method obtains state-of-the-art results on standard tracking benchmarks such as LaSOT, GOT-10k, AVisT. The authors further provide ablation studies and analyze the key components of the method.

**Strengths:**

1. The paper obtains strong results, outperforming recent trackers on large scale tracking benchmarks such as  LaSOT, GOT-10k, AVisT.
2. Rather than directly regress the target bounding box, or generate a score map indicating target location, the proposed architecture iteratively denoises the previous frame bounding box for tracking. This is novel and quite interesting. The method is also shown to perform very well.
3. The authors also employ a trajectory memory to integrate object motion information into the tracking which is shown to help. Most of the current trackers do not utilize any trajectory information, except using some weak prior that object motion is small. So it's interesting to see object trajectory directly utilized in the model prediction, rather than as a ad-hoc post-processing step.

**Weaknesses:**

1. While the authors present the method as a latent denoising approach motivated by DDPM, it is not fully clear how much of the DDPM theory is applicable to the method in the current form. For instance, how are each of the denoising stages supervised? Do you follow some noise schedule to supervise the output of each block? Unlike in a DDPM where a denoising network is shared and a timestamp is randomly sampled for each training iteration, the proposed approach utilizes a stacked architecture. Thus during training, are inputs to the i’th denoising block the outputs of i+1’th denoising block, or a randomly sampled noise? It would be helpful is the authors provide more details on how the intermediate denoising outputs are supervised.

2. I couldn’t find a clear experimental evidence on the benefit training the model to iteratively denoise the bounding box, instead of say directly predicting the total noise (i.e. the offset from previous to current box), or directly regressing the new box, while keeping the rest of the architecture identical. While the authors provide an ablation study in Table 5 on impact of denoising blocks, it seems to only analyze the impact of architectural choices in the denoising block (please correct me if I’m missing something here). I believe such a comparison is important the judge the impact of the core idea of doing iterative denoising.

3. There are some missing details for the training, especially related to training of visual and trajectory memory blocks. Do you perform a sequence based training to simulate the online tracking scenario? If not, how are the templates for the visual memory, as well as the trajectory obtained? Do you just use previous frame boxes (after adding some noise?), do you take consecutive frames from a video or sample randomly from a video?

**Questions:**

1. As stated in the weaknesss 1, please provide details on how the intermediate denoising outputs are supervised.

2. Related to weakness 2, what is the performance obtained by a model trained to directly predict the total noise, while keeping the rest of the architecture and training identical?

3. Please provide more details on the model training, as stated in weakness 3.

4. How many templates are stored in the visual memory? Is it 3? Do you always keep the first frame in the memory?

5. In table 4, when running the denoising process multiple times, how was the model trained? Do you just run the model corresponding to "Single forward pass denoising" multiple times, or train a new model in a DDPM style manner? Instead of using the same ViT-B architecture to do repeated denoising, what happens if you use a smaller model (ViT-S or say even small with only a couple of Transformer blocks), but apply it multiple times?

6. In table 5, what exactly does not doing"NoisePred" correspond to? What is the model architecture in this case?

7. It appears that the tracking performance consistently improves with using larger trajectory memory. Did you experiment beyond using 7 previous frames? Would using a larger trajectory cause significant latency overhead?

8. There seems to be a typo in line 114, in the definition of alpha.

9. Tracking datasets such as LaSOT can contain snippets with long occlusions. How does the model handle such cases? Do you simulate occlusions / out-of-frame scenarios during training to ensure that the denoising process doesn't diverge? In the limitations, the authors state that the trajectory memory can help to recover from occlusions. However since it contains only 7 previous frames, I'm not sure if that's sufficient to recover from such challenging cases since they can last longer. Does the visual memory help the tracker to redetect the object after long occlusions?

**Limitations:**

The authors briefly discuss the limitations.

---

> ### Author Rebuttal · Authors · 2024-08-07
>
> We thank the reviewer for the acknowledgement of the main contributions of our work, the useful comments and the relevant feedback provided on the technical side. We have provided detailed responses to your questions.
>
> >***Q1**.While the authors present the method as a latent denoising approach motivated by DDPM, it is not fully clear how much of the DDPM theory is applicable to the method in the current form.*
>
> - **Our method only borrows the noise addition technique from DDPM, but our training approach and denoising process are different from DDPM.** DDPM directly predicts noise during training, while we directly predict the ground truth. DDPM denoising requires sampling from a Gaussian distribution, which introduces randomness, whereas we can directly predict the target bounding box using a stacked denoising method. **Therefore, our method does not need to fully adhere to DDPM's theoretical framework.**
>
> - **Compared to DDPM, our method is more similar to the Denoising Autoencoder (DAE) [1].** A DAE adds noise to the input, passes it through an encoder to obtain the latent vector $z$, and then inputs $z$ into a decoder to reconstruct the output. In our method, we add noise to the bounding box and use a stacked denoising approach to obtain multiple intermediate latent vectors $z_i (i=1-12)$. We then input the latent vector $z_{12}$ into the box refining and mapping to reconstruct the target bounding box.
>
> -  **Below is a table that describes the differences between our method and DDPM, as well as the similarities with DAE:**
>
>    |  |DDPM|DeTrack| DAE|
>    |-|-|-|-|
>    |Noise addition|Gaussian noise|Gaussian noise|Gaussian noise|
>    | Input | Noisy image($x^*$) | Noisy box($b^*$) | Noisy image($x^*$) |
>    | Encoding | $z=f_\theta(x^*)$ |$z_{12}, z_{11}\cdots,z_{1}=f_\theta(b^*)$| $z=f_\theta(x^*)$ |
>    | Decoding | Noise $\epsilon_\theta=g_\theta(z)$ | box $b=g_\theta(z_{12})$ |image $x=g_\theta(z)$ |
>    |Optimization objective| $\epsilon-\epsilon_\theta$ | $b-b_\theta$|$x-x_\theta$|
>    |Inference| $x_{t-1} = \frac{1}{\sqrt{\alpha_t}}(x_{t}-\frac{\beta}{\sqrt{1-\bar{\alpha_t}}}\epsilon_\theta)+ \sigma_t \epsilon$ | $b=g(z_{12})$|$x=g(z)$|
>
> > ***Q2**.For instance, how are each of the denoising stages supervised?Do you follow some noise schedule to supervise the output of each block?*
>
> - We do not apply supervision to each block individually; instead, we use an implicit learning process similar to DAE[1]. By minimizing ||$b-b_\theta$||,the model is able to automatically learn the intermediate latent vectors.
>
> - We also experimented with adding supervision to intermediate layers, applying supervisory signals to each layer individually, but this approach did not yield better performance:
> || AO       |SR$_{0.5}$|SR$_{0.75}$|
> |-|-|-|-|
> |**Supervise each layer**|72.8|81.5|68.8|
> |**DeTrack**|**77.1**|**86.1**|**73.5**|
>
> - **We hypothesize that early supervision may make denoising more challenging.** For instance, adding supervision to the first layer implies that the model must complete the denoising in a single step. In contrast, applying supervision to the final layer allows the model to adaptively learn the denoising according different layers. As shown in Table 3 of the original paper, the denoising performance is relatively poor in the initial layers, and better denoising results are exhibited in the later layers.
>
> > ***Q3**.Thus during training, are inputs to the i’th denoising block the outputs of i+1’th denoising block, or a randomly sampled noise?*
> - The output of $i$th layer is the input of $i+1$th layer.
>
> > ***Q4**. I couldn’t find a clear experimental evidence on the benefit training the model to iteratively denoise the bounding box, instead of say directly predicting the total noise (i.e. the offset from previous to current box), or directly regressing the new box, while keeping the rest of the architecture identical.*
>
> - **Directly regressing the new box**. I apologize for any confusion. Please allow me to clarify. We have already conducted experiments where the bounding box is directly regressed while keeping the original structure unchanged, as shown in **Table 5 of the original paper**.
>
>    ||Deoising attentiion|NoisePred| AO       | SR$_{0.5}$ | SR$_{0.75}$ |
>    |-|-|-|-|-|-|
>    | **Directly regressing box(image)** ||| 74.0   | 84.7      | 72.5     |
>    | **Directly regressing box(box)** |✓|| 75.1    | 84.1       | 72.0      |
>    | **DeTrack**                    |✓|✓| **77.1** | **86.1**   | **73.5**    |||
>
> - The first row represents the case without any denoising module, directly regressing the bounding box from image features. The second row removes the noise prediction module,  similar to DAE[1], directly reconstructing the box to remove noise. It can be seen that DeTrack performs better, as DeTrack has learned the denoising pattern within the model's latent vector space.
>
> - **Predicting the total noise.** Based on your valuable suggestion, we conducted experiments predicting the total noise. The results are shown in the table below:
>
> - |                      | AO       | SR$_{0.5}$ |SR$_{0.75}$|
> | -------------------- | -------- | ---------- | -------- |
> | **Predicting the total noise** | 75.2  | 84.1     | 71.4  |
> | **DeTrack**          | **77.1** | **86.1**   | **73.5** |
>
> - Predicting the total noise resulted in a decrease of 1.9 in AO, 2 in SR0.5, and  2.1  in  SR0.75,compared to multiple noise predictions. We analyze that this is because predicting the total noise directly is more challenging than predicting it layer by layer. Layer-by-layer denoising allows the model to learn to filter out some noise at intermediate layers before arriving at the final result, rather than achieving it in one step.
>
> [1]. Vincent P, Larochelle H, Bengio Y, et al. Extracting and composing robust features with denoising autoencoders[C]//Proceedings of the 25th international conference on Machine learning. 2008: 1096-1103.

---

> ### Author Response · Authors · 2024-08-07
> **Further response serve to adress reiviewer's concerns（Part 1/2）**
>
> > ***Q5**. Do you perform a sequence based training to simulate the online tracking scenario? If not, how are the templates for the visual memory, as well as the trajectory obtained? Do you just use previous frame boxes (after adding some noise?), do you take consecutive frames from a video or sample randomly from a video?*
>
> - The training of our model is divided into three stages. In the first stage, there is no trajectory memory, only visual memory, which randomly samples two frames from the video. This is consistent with previous visual object tracking methods. In the second stage, trajectory memory is introduced, and sequential training is adopted. Consecutive frames are sampled from the video, with each frame's prediction result stored in the trajectory memory and updated in a first-in-first-out manner. The third stage involves training the IOU head. In this stage, the tracker is fixed and not trained; only the IOU head is trained to evaluate the quality of the template. **We greatly appreciate your suggestion, we will include these training details in the paper in the future， as it has helped us improve our manuscript further.**
>
> > ***Q6**. How many templates are stored in the visual memory? Is it 3? Do you always keep the first frame in the memory?*
> - The length of the visual memory bank is always 3 frames, which was experimentally verified to achieve the highest performance in Section 4.4.
> - The first frame is always fixed, as described in Section 3.3 on visual memory, where the first frame is fixed and only the dynamic frames are updated.
>
> > ***Q7**. in table 4, when running the denoising process multiple times, how was the model trained? Do you just run the model corresponding to "Single forward pass denoising" multiple times, or train a new model in a DDPM style manner? Instead of using the same ViT-B architecture to do repeated denoising, what happens if you use a smaller model (ViT-S or say even small with only a couple of Transformer blocks), but apply it multiple times?*
>
> - We run the model corresponding to "Single forward pass denoising" multiple times. Based on your suggestion, we retrained the model similarly to DDPM(**DiffusionTracking**). The results are shown in the table below:
> - |-|-|-|GOT-10k|-|-|-|
> |-|-|-|-|-|-|-|
> |**Denoising paradigm**|**Steps**|**AO**|**SR$_{0.5}$**|**SR$_{0.75}$**|**Flops**|**Speed**|
> |DiffusionTracking|1|71.8|81.0|69.6|120.0G|27fps|
> |DiffusionTracking|2|72.1|81.1|70.0|123.4G| 25fps |
> |DiffusionTracking|4|71.9|81.1|69.3|133.5|16fps|
> |DiffusionTracking|8|73.5|82.9|71.2|147.2G|14fps|
> |DiffusionTracking|12|72.5|81.9|70.2|**162.7G**|10fps|
> |**DeTrack**|12|**77.9**|**86.5**|**74.9**|119.0G|**30fps**|
>
> - |-|-|---|AVisT|---|---|TrackingNet|---|---|LaSOT|---|---|LaSOT$_{ext}$|---|Flops|Speed|
> |-|-|-|-|-|-|-|-|-|-|-|-|-|-|-|-|
> |**Denoising paradigm**|Steps|AUC|OP50|OP75|AUC|Pnorm|P|AUC|Pnorm|P|AUC|Pnorm|P|Flops|Speed|
> |DiffusionTracking|8|54.3|63.5|45.8|83.7|88.6|82.7|69.2|78.3|75.3|46.5|57.0|52.0|**147.1G**|14fps|
> |**DeTrack**|12|**60.2**|**69.1**|**50.2**|**84.5**|**89.1**|**83.6**|**72.9**|**81.7**|**79.1**|**53.6**|**64.4**|**50.4**|119.0G|**30fps**|
>
> - Based on your suggestion, we retrained the model with ViT-S, the table as show below:
>
> - | **Denoising paradigm**  | Steps | AO       | SR$_{0.5}$ | SR$_{0.75}$ |
> | ----------------------- | ----- | -------- | ---------- | ----------- |
> | **Recycle denoising**   | 96    | 68.9     | 78.2       | 63.2        |
> | **Recycle denoising**   | 48    | **69.4** | **78.5**   | **63.4**    |
> | **Recycle denoising**   | 24    | **69.4** | 78.0       | 63.0        |
> | **Single forward pass** | 12    | 69.1     | 78.3       | 62.9        |
>
> - It can be observed that there is a performance gap between ViT-S and ViT-B. This is because, in DeTrack, the ViT serves not only for denoising but also plays a crucial role in feature extraction and feature interaction. ViT-S, having a smaller model size, therefore has weaker representational capacity, and ViT-S lacks MAE pre-training, which is critical for tracking, as demonstrated in OSTrack[2]. Performance improves with multiple iterations of denoising on ViT-S, reaching its peak after 48 denoising steps.
>
> > ***Q8**. In table 5, what exactly does not doing"NoisePred" correspond to? What is the model architecture in this case?*
> - No NoisePred’ indicates the absence of a noise prediction module, where the model directly predicts the bounding box. The overall process is more similar to DAE [1]. For specific details, please refer to Q4.
>
> [1]. Vincent P, Larochelle H, Bengio Y, et al. Extracting and composing robust features with denoising autoencoders[C]//Proceedings of the 25th international conference on Machine learning. 2008: 1096-1103.

---

> ### Author Response · Authors · 2024-08-07
> **Further response serve to adress reiviewer's concerns（Part 2/2）**
>
> > ***Q9**. It appears that the tracking performance consistently improves with using larger trajectory memory. Did you experiment beyond using 7 previous frames? Would using a larger trajectory cause significant latency overhead?*
> - Based on your suggestion, we conducted experiments with a longer trajectory memory(On LaSOT dataset). The experimental results are as follows:
> - | **Length of trajectory memory**  |  AUC| Pnorm|P|
> | -| -------- | ---------- | ----------- |
> | **7**   | 71.3| **80.1**| 76.8 |
> | **8**   |71.3 | 79.8| 76.9  |
> | **9**   |70.7 | 79.0 | 76.0|
> | **10** | **71.6** |**80.1** |**77.2**|
> - Increasing the length of the trajectory memory indeed helps improve tracking performance. We will explore more details regarding trajectory memory length and include these findings in the paper. Thank you very much for your suggestion; it has been very helpful in enhancing our manuscript.
>
>
> > ***Q10**. There seems to be a typo in line 114, in the definition of alpha*
>
> - We will correct the typos in our subsequent revisions. Thank you very much for your suggestions, which are very helpful for improving our manuscript.
>
> > ***Q11**.Tracking datasets such as LaSOT can contain snippets with long occlusions. How does the model handle such cases? Do you simulate occlusions / out-of-frame scenarios during training to ensure that the denoising process doesn't diverge?*
>
> - Yes, during the training phase, we simulated out-of-view scenarios. Our bounding box coordinates are first normalized to the range -0.5 to 1.5, where the range 0-1 represents coordinates within the search area, and -0.5 to 0 and 1.0 to 1.5 represent coordinates outside the search area to simulate some out-of-view situations. I
>
> > ***Q12**. In the limitations, the authors state that the trajectory memory can help to recover from occlusions. However since it contains only 7 previous frames, I'm not sure if that's sufficient to recover from such challenging cases since they can last longer. Does the visual memory help the tracker to redetect the object after long occlusions?*
>
> - In the limitations section, we mention that our trajectory memory can recover the target in some cases, alleviating the out-of-view problem, but cannot completely solve it. This is a common issue with local trackers, including OSTrack[2], Mixformer[3], and others, and is also a limitation of our DeTrack. We will further investigate this issue in the future.
> - Visual memory also provides some benefit, as it can offer visual reference information.
>
> [1]. Vincent P, Larochelle H, Bengio Y, et al. Extracting and composing robust features with denoising autoencoders[C]//Proceedings of the 25th international conference on Machine learning. 2008: 1096-1103.
>
> [2] Ye B, Chang H, Ma B, et al. Joint feature learning and relation modeling for tracking: A one-stream framework[C]//European Conference on Computer Vision. Cham: Springer Nature Switzerland, 2022: 341-357.
>
> [3] Cui Y, Jiang C, Wang L, et al. Mixformer: End-to-end tracking with iterative mixed attention[C]//Proceedings of the IEEE/CVF conference on computer vision and pattern recognition. 2022: 13608-13618.

---

> > ### Comment · Reviewer_wkvN · 2024-08-09
> > **Official comment by reviewer wkvN**
> >
> > I thank the authors for providing a detailed rebuttal. It answers most of my concerns. Hence I'm raising my rating to "Accept".
> >
> > I would suggest that the authors try to incorporate the additional details provided in the rebuttal regarding the denoising process, sequences training etc in the manuscript since this will be very helpful. I think it would also be better to motivate the method from the point of denoising autoencoders rather than DDPM. Additionally I would suggest to tone down on the claims of "single forward pass". A standard diffusion model can also be viewed as a "single forward pass" model which just stacks multiple blocks with shared weights.
> >
> > One additional question: The authors state that they do not provide any auxiliary supervision to the intermediate denoising stages. Thus when generating the results in Table 3 or Figure 4, are you just reusing the final "mapping" block to generate the intermediate boxes (i.e. after steps 1, 2, 3 ...) from the denoised latents?

---

> > > ### Author Response · Authors · 2024-08-09
> > >
> > > Thank you very much for your response and for recognizing our work.
> > > > ***Q1**.  When generating the results in Table 3 or Figure 4, are you just reusing the final "mapping" block to generate the intermediate boxes (i.e. after steps 1, 2, 3 ...) from the denoised latents?*
> > >
> > > - Yes, we use the mapping layer to project the denoised boxes into the image coordinate space. I will include these details in the paper to help improve the quality of the manuscript. Thank you very much for your question.
> > >
> > > > ***Q2**. I would suggest that the authors try to incorporate the additional details provided in the rebuttal regarding the denoising process, sequences training etc in the manuscript since this will be very helpful.*
> > >
> > > - Based on your suggestions, we will incorporate the additional details regarding the denoising process, sequence training, and other relevant aspects into the manuscript, as this will greatly enhance the quality of our paper.
> > >
> > > >***Q3**.  I think it would also be better to motivate the method from the point of denoising autoencoders rather than DDPM.*
> > > - We will also describe our method from the perspective of denoising autoencoders rather than DDPM.
> > >
> > > > ***Q4**. I would suggest to tone down on the claims of "single forward pass".*
> > > - Additionally, we will reduce the emphasis on the concept of a single forward pass and instead focus on highlighting our denoising model.

---

### Official Review · Reviewer_JcZZ · 2024-07-12

**Soundness:** 3
**Presentation:** 3
**Contribution:** 2
**Rating:** 7
**Confidence:** 5

**Summary:**

In this paper, the authors reformulate the visual tracking problem using denoising learning process. Given noisy bounding box coordinates, the conditional denoising tracker model's task is to remove the bounding box noise for accurate target state estimation. For the denoising formulation, they introduce an in-model latent denoising process where they claim that it can be executed with a single forward pass of the model. The effectiveness of their proposed method is validated on various visual tracking datasets.

**Strengths:**

- The authors evaluate their proposed method on multiple challenging visual tracking benchmarks, where they show competitive performance on these datasets to some extent when compared to other state-of-the-art methods.

- The authors show results for multiple ablation experiments on varying hyperparameter settings such as memory length and threshold value, showing some robustness to hyperparameter choices.

- The motivation of the proposed method is plausible, where the authors focus on the timely subject of reformulating visual tracking to solve the persisting issues of previous two-stream feature matching-based visual tracking methods.

**Weaknesses:**

- The claim that the model only require a "single forward pass" seems somewhat misleading since the full latent denoising model contains $l=12$ denoising blocks and require same number of denoising steps for full inference.

- From my understanding, the motivation for generative models that employ iterative denoising process using Markov chains was that modeling the complex data distribution with multiple modes cannot be achieved with a single Gaussian normal distribution. From this perspective, what is the motivation for modeling visual tracking using this iterative multi-step process, since it seems overly cumbersome and intricate.

- Related to the previous question, how does changing the number of denoising blocks $l$ affect the performance of the tracker? Experimental results in Table 3 does not seem enough since the model is trained with $l=12$ as its default value.

**Questions:**

Please refer to the questions in the weaknesses section.

**Limitations:**

The authors included a separate discussions and limitations section in their paper, with adequate explanations and descriptions on the weaknesses and possible future directions.

---

> ### Author Rebuttal · Authors · 2024-08-07
>
> We extend our heartfelt gratitude for your perceptive insights and your recognition of the unique and meaningful contributions made by our research. Your support is highly valued, and we would be honored if you could serve as an advocate for our work.
>
> >***Q1**.The claim that the model only require a "single forward pass" seems somewhat misleading since the full latent denoising model contains 𝑙=12 denoising blocks and require same number of denoising steps for full inference.*
>
> - I apologize for the confusion. While it is true that we perform denoising in 12 steps, these 12 denoising steps are included  within the Denoising ViT. Therefore, What we intend to convey is not that the denoising steps require a single forward pass, but rather that our method requires only **a single forward pass through the entire network** to complete the denoising process. In contrast, diffusion models require **multiple forward passes through the entire network** to achieve denoising. The specific differences are as follows:
> | ------------------------------------------------------------------------------------- | Diffusion Model | DeTrack |
> | ------------------------------------------------------------ | --------------- | ------- |
> | **Number of forward passes through the entire network**      | n times         | 1 time  |
> Therefore, we refer to it as a "single forward pass."
>
> >***Q2**. What is the motivation for modeling visual tracking using this iterative multi-step process?*
>
> - **Noisy bounding boxes, which can have arbitrary shapes and positions.** Using training set bounding boxes may not  provide sufficient robustness to unseen data. By introducing Gaussian noise to create noisy boxes with arbitrary shapes and positions, we enhance robustness during testing. On GOT-10k, where training and testing sets have different categories, our method shows superior performance, demonstrating that adding noise effectively improves the tracker's robustness to unknown data.
>
> - **From frame t−1 to frame t, we assume that the object undergoes numerous small movements.** These movements are complexly distribution in nature, with each step's movement being dependent on the previous position. Therefore, we model the object's motion pattern using this approach. We simulate these numerous small movements with Gaussian noise, where the process from frame t to frame t−1 is viewed as the noise-adding process, and from frame t−1 to frame t as the denoising process.
>
> >***Q3**.Related to the previous question, how does changing the number of denoising blocks 𝑙 affect the performance of the tracker? Experimental results in Table 3 does not seem enough since the model is trained with 𝑙=12 as its default value.*
>
> -  Based on your valuble suggestion, we retrained the model according your suggestion. We added supervision signals at different layers and changed the number $l$ of denoising blocks to evaluate their performance impact on the tracker. The latest results are shown in the table below: compared to Table 3 in the original paper, the updated results demonstrate higher accuracy in the first 8 steps, with faster convergence in denoising. However, by the 10th step, the performance matches that in Table 3, and at the 11th step, it is even slightly lower than in Table 3.
> ||step1|step2|step3|step4|step5|step6|step7|step8|step9|step10|step11|step12|
> |-|-|-|-|-|-|-|-|-|-|-|-|-|
> |AO|16.9|21.9|25.2|34.8|42.5|48.7|52.4|60.6|65.4|70.5|71.6|77.1|
> |SR$_{0.5}$|12.5|17.2|21.3|34.9|43.1|52.3|57.3|67.1|72.4|78.2|80.8|86.1|
> |SR$_{0.75}$|2.9|6.9|9.7|16.9|23.9|32.1|37.3|49.1|55.5|62.6|65.9|73.5|
> We greatly appreciate your suggestion, as it has helped us improve our manuscript further.

---

> > ### Comment · Reviewer_JcZZ · 2024-08-11
> >
> > I thank the authors for their detailed response to my questions, and the additional details provided by the authors were very helpful.
> > Also, based on the concerns from the other reviewers and the authors' response to address these issues, I am inclined to raise my rating to "accept".
> > However, aside from the novel approach proposed in the manuscript, I also suggest the authors to reconsider their claim that the proposed algorithm requires only "single forward pass", since it can confuse the readers.

---

> > > ### Author Response · Authors · 2024-08-12
> > >
> > > We sincerely appreciate the reviewer's response and are very grateful for the reviewer's recognition of our work. Additionally, we will no longer claim that our model requires a single forward pass. Your suggestion is very helpful and will aid us in improving the manuscript.

---

### Author Rebuttal · Authors · 2024-08-07

We sincerely appreciate the thorough review provided by all the reviewers. The valuable feedback from the reviewers has significantly contributed to enhancing the quality of our manuscript. We extend our gratitude to Reviewer **JcZZ**, Reviewer **wkvN** and Reviewer **b9Bg** for acknowledging the novelty of our work. Their positive recognition of the innovation in our research is greatly appreciated.
Based on the comments from the reviewers, I have summarized the strengths of our paper as follows:

- Show competitive performance on these datasets to some extent when compared to other state-of-the-art methods.(Reviewer **JcZZ**)
- Rather than directly regress the target bounding box, or generate a score map indicating target location, the proposed architecture iteratively denoises the previous frame bounding box for tracking. This is novel and quite interesting. The method is also shown to perform very well. (Reviewer **b9Bg**)
- The idea of denoising learning for tracking is novel and well-motivated, and the proposed pipeline is streamlined and promising to lead a new tracking paradigm.(Reviewer **wkvN**)

We have summarized our novelty as follows:

- We propose a novel in-model latent denoising learning paradigm for visual object tracking, which provides a new perspective for the research community. It decomposes the classical explicit recycle denosing process into several denoising blocks and solves the problem with a tracking network in a single forward pass, which is valuable for real applications.

- We present a tracking model including a denoising ViT, comprised of multiple denoising blocks. The denoising process can be completed by progressively denoising through the denoising blocks within ViT. Furthermore, we construct a compound memory in the model that improve the tracking results using visual features and trajectory.

- Experimental results on several popular experiments, including AVisT, GOT-10k, LaSOT, and LaSOT$_{ext}$,  demonstrate that the proposed method achieve competitive results.

We hold the belief that these innovative contributions elevate the value and significance of our research in the area of visual object tracking.

We intend to heed the insightful suggestions from the reviewers by incorporating additional essential experiments. Additionally, a thorough review of the manuscript will be conducted to rectify any typographical and grammatical errors.

---

### Decision · Program_Chairs · 2024-09-25

**Decision:**

Accept (poster)

**Comment:**

After rebuttal and discussion, all reviewers agree that this manuscript should be accepted: two plain accepts and one weak accept. The main strengths of the manuscript is the novelty of the method, the good performance, and the reproducibility.